# Viral Metagenomic Next-Generation Sequencing for One Health Discovery and Surveillance of (Re)Emerging Viruses: A Deep Review

**DOI:** 10.3390/ijms26199831

**Published:** 2025-10-09

**Authors:** Tristan Russell, Elisa Formiconi, Mícheál Casey, Maíre McElroy, Patrick W. G. Mallon, Virginie W. Gautier

**Affiliations:** 1UCD Centre for Experimental Pathogen Host Research (CEPHR), University College Dublin, D04 E1W1 Dublin, Ireland; 2UCD School of Medicine, University College Dublin, D04 C7X2 Dublin, Ireland; 3Regional Veterinary Laboratories (RVL) Division, Department of Agriculture, Food and the Marine, Agriculture House, Backweston, T12 XD51 Dublin, Ireland; 4Department of Agriculture, Food, and the Marine Laboratories, Backweston, Celbridge, W23 VW2C Kildare, Ireland; 5Department of Infectious Diseases, St Vincent’s University Hospital, D04 T6F4 Dublin, Ireland; 6UCD Conway Institute, University College Dublin, D04 E1W1 Dublin, Ireland

**Keywords:** (re)emerging viruses, metagenomic next-generation sequencing, one health, surveillance, pandemic preparedness

## Abstract

Viral metagenomic next-generation sequencing (vmNGS) has transformed our capacity for the untargeted detection and characterisation of (re)emerging zoonotic viruses, surpassing the limitations of traditional targeted diagnostics. In this review, we critically evaluate the current landscape of vmNGS, highlighting its integration within the One Health paradigm and its application to the surveillance and discovery of (re)emerging viruses at the human–animal–environment interface. We provide a detailed overview of vmNGS workflows including sample selection, nucleic acid extraction, host depletion, virus enrichment, sequencing platforms, and bioinformatic pipelines, all tailored to maximise sensitivity and specificity for diverse sample types. Through selected case studies, including SARS-CoV-2, mpox, Zika virus, and a novel henipavirus, we illustrate the impact of vmNGS in outbreak detection, genomic surveillance, molecular epidemiology, and the development of diagnostics and vaccines. The review further examines the relative strengths and limitations of vmNGS in both passive and active surveillance, addressing barriers such as cost, infrastructure requirements, and the need for interdisciplinary collaboration. By integrating molecular, ecological, and public health perspectives, vmNGS stands as a central tool for early warning, comprehensive monitoring, and informed intervention against (re)emerging viral threats, underscoring its critical role in global pandemic preparedness and zoonotic disease control.

## 1. Introduction

The millions of fatalities and cases of morbidity resulting from the COVID-19, 2009/2010 swine flu, and AIDS pandemics, along with recurrent outbreaks of flaviviruses, filoviruses, poxviruses, and others, illustrates the ongoing threat posed by (re)emerging viruses to human health [1]. The next pandemic may arise from an entirely unknown source, so-called Disease X in humans or Disease Y in animals, or rapidly evolving pathogens. Multiple recent outbreaks of novel aetiology have been caused by the zoonotic spillover of viruses from animals to humans, which is driven by changes in animal, environmental, and human health, aligning the Disease X/Y issue with the One Health paradigm introduced in Section 1.2. Timely detection and strategic surveillance of Disease X/Y causing viruses are critical components of pandemic preparedness, but the suitability of current PCR diagnostics is limited by the lack of genetic references. In these scenarios, untargeted approaches that do not require prior sequence knowledge such as metagenomic next-generation sequencing (mNGS) have become increasingly valuable for both virus discovery and genomic surveillance. This review examines the current literature using mNGS for virus discovery and surveillance to propose a practical framework for integrating mNGS into (re)emerging virus surveillance strategies.

### 1.1. Metagenomic Next-Generation Sequencing: Transforming Virus Discovery and Surveillance

mNGS refers to the untargeted sequencing of all DNA and/or RNA present in a sample, the metagenome, enabling comprehensive identification of the diverse organisms from which nucleic acids originate. The integration of mNGS into pathogen discovery and surveillance is the culmination of technological advancements in molecular biology and NGS over the last 50 years (Figure 1).

Unlike first-generation sequencing, pioneered by Fred Sanger in 1977 [18], second- and third-generation sequencing platforms generate much larger quantities of data and deliver whole genome sequences more rapidly and at substantially reduced costs. Critically, mNGS is sequence-independent, allowing for unbiased detection of unknown or unexpected pathogens (Table 1).

The evolution of sequencing technologies is illustrated by comparing the discovery of three major zoonotic coronaviruses that have emerged in humans during the 21st century (Figure 1). In 2002/2003, SARS-CoV-1 was identified as the aetiological cause of a pneumonia epidemic using a combination of virus isolation, electron microscopy, serology, histology, PCR, and partial genome sequencing via Sanger technology [10,11,22]. A decade later, the identification of MERS-CoV in 2012 leveraged similar methods but also incorporated whole genome sequencing (WGS) using viral enrichment methods, such as nuclease digestion and filtering as described in Section 2.2, and the Roche 454 short-read NGS platform [15]. More recently, in 2019, SARS-CoV-2 was directly identified from patient samples using short-read mNGS with the Illumina platform producing a complete viral genome sequence within days [16].

Continuous advancements in NGS technologies have greatly accelerated and broadened their practical applications, and their use in virus identification. As these technologies become more accessible, routine use of unbiased sequencing is poised to become central to rapid pathogen discovery and surveillance in the near future (Figure 1).

### 1.2. Pathogen Discovery and Surveillance: One Health Imperative

Approximately 60–80% of (re)emerging human viruses have zoonotic origins or circulate frequently between humans and animals [23,24,25,26]. Zoonosis, the transmission of a pathogen from non-human animals to humans [27], can also occur in the other direction (reverse zoonosis) and between different non-human species. Many (re)emerging threats are arboviruses, transmitted by arthropod vectors, which maintain both sylvatic (wildlife) and urban (domestic animal) cycles, before infecting people [28].

The ability of viruses to adapt to new host species is a key driver of their emergence in humans, as seen with SARS-CoV-1, where the virus evolved in civet cats (intermediate host) after spillover from bats (reservoir host), thus facilitating human SARS-CoV-1 infections [29]. Concerns were raised when cattle–cattle transmission of the highly pathogenic avian influenza virus H5N1 (HPAIV) was observed in 2024–2025, as this could indicate a lowering of the evolutionary barrier to sustained human–human transmission [30,31]. Currently humans are dead-end hosts of HPAIV because the virus is not transmitted from an infected human to any other species. Sequence-based surveillance of pathogens in animal populations enables early detection of mutations that may elevate zoonotic risk and helps prepare rapid response strategies.

The emergence of infectious diseases is driven by a complex interplay of human activities and environmental changes that intensify contact between species. Climate change driven by human activities, including deforestation, intensive farming, and encroachment on natural habitats, can cause biodiversity loss, disrupt ecosystems, and facilitate spillover events. Demographic trends such as urbanisation, population growth, and ageing populations, along with globalisation via travel and trade, accelerate virus dissemination. Following a spillover event, viruses can rapidly disseminate in immunologically naïve populations, with far reaching implications for conservation, agriculture, food security, and human health [32,33,34].

These multidimensional challenges reinforce the importance of the One Health paradigm, which recognises the interdependence of animal, environmental, and human health (Figure 2). Coined by wildlife researchers [35], the “One Health” framework calls for interdisciplinary and cross-sectoral collaboration to generate equitable, sustainable, and global solutions to emerging health threats. The World Health Organisation (WHO) Pandemic Agreement, adopted by WHO Member States in May 2025, is designed to strengthen global cooperation to promote the holistic One Health approach (Figure 2) to prevent or respond to future pandemics [36]. A central element of this agenda is enhanced surveillance of known or high-risk zoonotic and arboviral pathogens in animals and the environment. Regional authorities such as the EU also offer financial support to member states implementing One Health approaches, with initiatives such as EU4Health funding research and programmes that strengthen pandemic preparedness through, for example, surveillance [37].

In this context, surveillance involves the monitoring and tracking of pathogens as they (re)emerge and spread to new locations. This requires pathogen identification, for which vmNGS can be used; knowledge of where and from whom the pathogen was isolated; and analysis to determine the relatedness of the identified pathogen to previous isolates. There are two forms of surveillance: active, involving the proactive detection of pathogens before they cause disease in the at-risk populations, and passive, involving the responsive detection of pathogens after they cause disease.

This review focuses on the capabilities and applications of viral mNGS as a uniquely versatile tool for One Health pathogen discovery and surveillance. Its sequence-independent methodology enables rapid discovery of unknown pathogens, so-called Pathogen X or Y, without the need for prior genetic information, making it invaluable for outbreak investigations. At the same time, mNGS supports comprehensive viral genome surveillance, enabling real-time monitoring of viral evolution, identification of origins, and tracking of dissemination routes. To begin, the technical and practical considerations of vmNGS workflows are addressed by detailing each step from sample collection to nucleic acid extraction, sequencing, and computational identification. The capabilities and limitations of vmNGS are then illustrated with practical examples and real-world applications, highlighting its deployment in the surveillance and discovery of zoonotic viruses for environmental and animal monitoring, and investigations of infectious cases of unknown aetiology. Finally, this review synthesises these insights to propose an integrated framework for effective emerging-pathogen surveillance, focusing on the prospective role of vmNGS in One Health Strategies.

## 2. Viral Metagenomic Next-Generation Sequencing Workflows

Key steps of vmNGS workflows include sampling, extraction, virus enrichment, library preparation, sequencing, and bioinformatic analysis, which must be tailored to specific surveillance goals and sample matrices (Figure 3). Samples such as tissues, swabs, fluids, air, and dust demand distinct approaches due to variability in viral abundance, viral particle integrity, viral nucleic acid concentration, and matrix composition (see Section 2.1). Clinical specimens typically follow optimised tissue or swab/fluid workflows, while environmental samples often require additional enrichment strategies to compensate for low viral loads (see Section 2.2). The two primary NGS platforms are short-read Illumina sequencing and long-read Oxford Nanopore Technologies (ONT) sequencing (see Section 2.3). Downstream, bioinformatic pipelines are essential for quality control (QC), contaminant removal, assembly of overlapping reads into contigs, and taxonomic identification (see Section 2.4). Quality assurance and standardisation of vmNGS workflows are necessary for inter-study comparisons and reliability of results (see Section 2.5).

### 2.1. Strategic Sample Selection and Quality Control in vmNGS

Thoughtful and targeted sample selection is fundamental to the success of vmNGS, directly influencing NA extraction and enrichment strategies, downstream processing, sequencing efficiency, and the accuracy of pathogen detection. Whether in clinical or environmental surveillance, the choice of sample type and site collection should be informed by the likely site of pathogen presence, the properties of the sampling matrix, and relevant information such as disease symptoms or weather conditions at the time of sample collection.

#### 2.1.1. Environmental Settings

In environmental surveillance, the location of sample collection significantly increases the likelihood of detecting circulating pathogens, thereby improving both sensitivity and cost-efficiency. For example, collecting water samples near agricultural, industrial, or urban settings increases the likelihood of detecting a circulating pathogen, as shown by the positive correlation between polio detection and population size for samples collected from multiple wastewater sites in Nigeria [38], while air and dust samples should be collected from densely populated indoor environments [39]. Pathogen detection during environmental surveillance is also associated with seasonal fluctuations [39]. Working within the available resources is also challenging, especially when using mNGS as the detection tool, so periodic untargeted pathogen detection could be used to guide routine targeted environmental surveillance, which normally has reduced overall costs.

#### 2.1.2. Clinical and Postmortem Context

Necropsy findings, clinical signs, and reported symptoms should guide clinical sample selection to improve the likelihood of detecting the aetiological agents by vmNGS. For respiratory illness, nasopharyngeal swabs are appropriate, while cerebral spinal fluid (CSF) is most informative in cases of suspected neurological infection. Faecal samples or cloacal swabs are optimal for gastrointestinal diseases, and suspected arbovirus infections support collection of blood samples. Evidence-based sample selection had a direct impact on sensitivity during the 2024 HPAIV H5N1 outbreak in cattle in the USA, where only 10% of nasal swabs, serum, and blood buffy coat tested positive [31]. Subsequent analysis of milk samples, which aligned with clinical signs of mastitis, increased detection rate to 80%, outlining the importance of logically matching sample collection with symptom presentation [31]. When symptoms are non-specific or when suitable samples are inaccessible, tissues or fluids with a high likelihood of containing the pathogen, such as the spleen or blood, can be prioritised due to their roles in immunity and systemic infection. Equally important is the timing, as virus abundance tends to peak during the acute phase of infection, and late sample collection could miss the pathogen due to immune clearance.

### 2.2. Increasing Sensitivity with Host Depletion and Virus Enrichment

vmNGS faces sensitivity challenges due to the low abundance of viral nucleic acid amidst high levels of host and environmental genetic material in most clinical and environmental samples. To address this, various strategies have been developed to increase the proportion of viral nucleic acid within the total nucleic acid (TNA) pool at different stages: pre-extraction, during extraction, or during library preparation. These methods broadly fall into two categories: depletion of non-viral material and targeted virus enrichment (Figure 3).

#### 2.2.1. Depletion of Non-Viral Material

Depletion of non-viral material focuses on selectively removing or excluding animal, plant, bacterial, fungal, and parasitic material from the sample (Figure 3). Pre-extraction methods, including low-speed centrifugation and filtration, take advantage of the smaller size and lower density of viral particles compared to eukaryotic and prokaryotic cells, enabling separation of virions from larger and denser components in biological fluids, swab fluid, and some environmental samples.

Other depletion strategies are based on differences in viral and non-viral nucleic acid (Figure 3). Nuclease treatments, for instance, degrade unprotected host and bacterial nucleic acids, sparing DNA and RNA shielded within viral capsids [40,41,42,43]. Improved sensitivity in influenza virus detection was achieved using nuclease treatment to increase the fraction of viral reads from 0.046% to 6.04% [14]. However, these methods depend on sample integrity and have been shown to digest viral genomic DNA, which often lacks the protective capsid [44]. Antibody-based removal of methylated DNA can also be used, because viral DNA is not methylated [45,46,47]. Post-extraction ribosomal RNA (rRNA) depletion is especially relevant as rRNA typically makes up over 90% of total RNA extracted from tissues [40,43,48,49,50,51,52]. Commercial rRNA depletion kits often use probes designed against human and rodent rRNA sequences for enzymatic or affinity-based depletion, but some kits are effective for avian rRNA causing a 700-fold reduction as well [26,27]. Tissue sample TNA extracts are often treated with DNase, assuming actively replicating DNA and RNA viruses transcribe their genes and should be detected using a transcriptomics approach [40].

A balance must be found between enriching for viral nucleic acids and over-processing, because the strategies described here can cause off-target depletion of viral nucleic acids and RNA degradation, which reduces sensitivity and generates false negatives [53]. Very low TNA concentrations resulting from host depletion often necessitate untargeted amplification to meet the minimum input requirement for library preparation and sequencing platforms (Box 1).

Box 1Sequence-independent amplification.Swabs and biological fluids often
contain very low TNA concentrations following host depletion, so
amplification of nucleic acid is required to increase the concentration. Many
NGS library preparation methods and technologies have a minimal input of starting
DNA/RNA. Sequence-independent, single-primer amplification (SISPA)
non-specifically amplifies DNA or cDNA derived from RNA using a primer with
6–10 random bases at its 3’ end and a known sequence at its 5’ end, so PCR
using a primer against the known sequence can be carried out [54]. This has been used for Illumina [55] and ONT NGS platforms (see Section 2.3) [56,57]. SISPA has also been modified to incorporate adapter sequences
required for the Illumina flow cells, which avoids the need for expensive
commercial library preparation kits [58].

#### 2.2.2. Virus Enrichment

Hybrid capture (HC) is one form of viral nucleic acid enrichment, which uses biotinylated probes or probes conjugated to magnetic beads designed to bind sequences from viral families of interest. Following probe hybridisation, either pre- or post-library preparation, bound nucleic acids are captured while unbound material is washed away. If required, sequence-independent amplification can be performed to increase the input amounts before library preparation (Box 1).

Studies using artificially spiked samples have shown HC dramatically increases the sensitivity of vmNGS. The lower limit of detection (LOD) for HC-vmNGS can reach as low as 10 virus particles/mL, compared to 10^3^–10^4^ virus particles/mL for untargeted approaches [34,35,36]. Similarly, genome coverage can exceed 90% with HC compared to less than 50% without [59,60,61]. Real-world applications have confirmed these advantages in various sample types, including plasma, respiratory secretions, and environmental matrices [59,60,62,63,64]. Notably, highly sensitive detection of zoonotic viruses such as *Orthohantavirus* and *Coronaviridae* in both animal and human samples has been achieved using custom enrichment probe sets [65,66]. In One Health surveillance studies, custom probe sets targeting 663 viral species identified 27 species of 11 families in bovine rectal swabs compared to 11 species from 6 families without HC [61].

Despite clear sensitivity benefits, HC-vmNGS has some limitations, such as reduced performance with degraded samples due to impaired probe binding [66]. The method effectiveness is also reduced for viruses not represented in the probe set, making it less suitable for pathogen discovery [67]. Lists of known viral species are growing, allowing HC probe sets to be expanded. However, generating or purchasing probe sets can be expensive, although this can be offset by increased sample pooling per sequencing run thanks to reduced depth requirements. Using polyethylene glycol (PEG) to precipitate viral particles and then forming a pellet with high-speed centrifugation can also be used to enrich the virus without the need for probe sets [68,69,70,71,72]. The effectiveness of PEG-vmNGS is dependent on intact viral particles, limiting its use, while a trade-off shared by HC- and PEG-vmNGS compared to untargeted vmNGS is the loss of dataset richness for non-viral, microbial, and host identification; antimicrobial resistance gene surveillance; or host transcriptomic studies [26]. Nevertheless, for samples containing low concentrations of viral particles and nucleic acids, such as wastewater, the increased sensitivity of HC- or PEG-vmNGS is highly valuable. For broader pathogen discovery or characterisation of complex microbiomes, untargeted vmNGS remains the preferred strategy.

### 2.3. Next-Generation Sequencing Technologies

NGS technologies enable comprehensive analyses of nucleic acids from complex samples by generating millions of nucleic acid sequence reads, which can be compared computationally to reference databases for pathogen identification (see Section 2.4). NGS workflows begin with the preparation of sequencing libraries from DNA and RNA extracts. This process converts extracted nucleic acid into a format compatible with the NGS platform, typically involving fragmentation, adapter ligation, and incorporation of the barcode for multiplexing.

The Illumina instrument produces large volumes of short reads, typically in the range of 50–300 bp long (Table 1) [73]. To account for this read length, input DNA or cDNA must be fragmented during library preparation with fragmentation tailored to the nucleic acid integrity of the sample. Without adequate fragmentation, only the terminal fragments of long nucleic acids would be sequenced. ONT platforms permit sequencing of long reads reaching lengths of 10–1000 Kb with no strict upper limits (Table 1) [73]. This eliminates the need for fragmentation during library preparation. Both technologies require the ligation of adapter sequences to the ends of nucleic acids so they hybridise to flow cells. When multiplexing samples on a single flow cell, unique sequence barcodes are incorporated to differentiate sequencing reads and facilitate sample identification post-run.

Short-read sequencing, also known as shotgun sequencing, normally uses the fluorophore-based Illumina technology, while ONT uses the disruption of electronic fields as nucleic acid strands move through a pore to achieve long-reads (Table 1). Recent advances in ONT and Illumina technologies mean the latest instruments can produce terabytes of data per run. Illumina can yield billions of short reads, while ONT produces millions of long reads. Increased sensitivity would be expected due to the increased depth from Illumina, but in practice the two technologies have similar sensitivity and ONT achieves greater genome coverage [74,75,76,77]. Additional advantages of ONT include the portability of its Minion instrument, real-time sequence analysis and adaptive sequencing where reads corresponding to a predefined target are included and off-target reads are excluded in real-time (Table 1). Illumina is advantageous when working with degraded or low-yield samples and has greater accuracy, as is required for applications such as SNP detection or strain/variant identification (Table 1).

### 2.4. Bioinformatic Analysis for Virus Discovery, Characterisation, and Molecular Epidemiology

The gigabytes or terabytes of data generated by an NGS run requires advanced computational methods for effective downstream processing of sequence data and meaningful interpretation in the context of complex clinical or environmental samples. Bioinformatic analysis is essential for both the identification and the molecular epidemiology of emerging and circulating viruses.

This process includes several interlinked steps starting with quality control to filter out low-quality, contaminant, and host reads and de novo assembly of continuous sequences (contigs) based on read overlaps, a step particularly critical for short-read platforms but less so for long-read systems like ONT. These assembled sequences or direct reads are then searched against reference databases to assign taxonomy and identify pathogens (Figure 4). Software packages incorporating these tools have been developed for streamlining and ease-of-use [78,79]. Specific tools have been developed for de novo assembly of reads obtained by mNGS [80] and vmNGS [81] enabling genome assembly of novel or highly variable viral genomes. More in-depth comparisons of bioinformatic tools for vmNGS can be found in these recent reviews [82,83].

A lack of suitable reference genomes for newly discovered viruses and the sheer quantity of data generated by modern NGS instruments has transformed the bioinformatic landscape. As such, artificial intelligence (AI) tools are increasingly being developed and utilised for computationally intensive processes such as taxonomic identification, genome resolution, and annotation [84]. Modern AI tools have already proven effective in mining existing mNGS datasets to expand the virosphere [85]. Limitations around sensitivity exist compared to more-established tools, with a thorough review of the advantages and disadvantages of AI-based tools published recently [84].

After sequence assignment and pathogen identification, molecular epidemiological analyses are performed to understand patterns of variations, signature of host adaptation, and molecular evolution (Figure 4). When necessary, tiled amplicon and NGS can be integrated with WGS workflows to ensure robust genome coverage and sensitivity. Phylogenetic analysis, which examines the genetic relationships and evolution of viral lineages is the cornerstone of molecular epidemiology. The rich genetic information and sequence variation obtained facilitate in-depth tracing of outbreak origins, transmission pathways, and evolutionary trends and inform public health intervention [86].

Functional analysis and viral gene annotation is achieved using homology-based computational approaches to match new sequences with known gene functions, while powerful AI tools can be used to model protein structures given genetic sequences [87], then predict host–virus protein–protein interactions [88].

The breadth and depth of information derived from vmNGS bioinformatics demands significant expertise, computing power, and data storage infrastructure and space, which presents both opportunities and challenges for maximising the impact of mNGS in viral discovery and molecular surveillance.

### 2.5. Maintaining Quality Control and Standardisation of vmNGS

QC is vital throughout vmNGS with standardisation of protocols and quality assurance metrics used for assessing workflow performance. Sample and nucleic acid integrity play critical roles in workflow selection and success. Highly degraded samples may limit the applicability of certain enrichment or sequencing approaches, such as host depletion methods (see Section 2.2) that rely on intact viral particles and some long-read NGS technologies (see Section 2.3). There is greater flexibility and more options available when sample and nucleic acid integrity is maintained, which begins with strict adherence to standard operating procedures for collection, transport and storage (see Box 2).

Sequencing the metagenome can detect all nucleic acids, so contamination poses a significant risk to specificity. Contamination can arise externally from the laboratory environment, such as from personnel or sampling equipment, and internally, such as from cross-contamination between samples. While complete avoidance of contamination is not possible, it can be minimised by implementing standard operating procedures (see Box 2) and tracked by including negative controls [89].

Box 2Sample handling, shipping, and storage.Nucleic acids and viral particles in clinical and
environmental samples can be degraded by long term storage at non-freezing
temperatures, acidic and alkaline pH, and freeze–thaw cycles [90,91]. Suitable storage buffers and maintaining the cold chain
protects nucleic acids and viruses from degradation and loss of viability,
respectively, while short-term storage (<1 week) at 4 °C avoids
freeze–thaw cycles. Appropriate buffers used for sample storage depend on the
intended outcomes. RNA is especially sensitive to degradation by RNases so
samples should immediately be stored at <4 °C to reduce RNase activity or
stored in buffers containing inhibitors of RNases, such as guanidium
thiocyanate, which also inactivates virus [92]. If shipping samples
across borders or virus isolation is not required, molecular transport media
or lysis buffers from commercial RNA extraction kits can be used to
inactivate viruses while protecting nucleic acids [92]. Universal or virus transport media can be used if
viable virus is required [90].

Consistent technical validity and comparability of vmNGS workflows rely on harmonised QC metrics, robust proficiency testing, and precise documentation of technical parameters from sample acquisition through bioinformatic analysis. Unlike PCR, vmNGS lacks a targeted amplification step, making its LOD highly dependent on sequencing depth and sample integrity, as only a small fraction of total reads may identify low-abundance viral genomes in complex matrices or degraded samples, thus requiring high coverage and increased costs [93,94,95]. Key determinants of sensitivity and reliability include flow cell capacity, careful selection and validation of sample and library quality, bioinformatic strategies, and the use of pooling strategies with unique molecular identifiers, which can improve efficiency but must be balanced against the risks of insufficient per-sample depth when batching multiple samples. A rigorous approach entails matching sample complexity and expected pathogen abundance to required depth, and validating vmNGS workflows with reference samples with known pathogen status or spiked-in standards for benchmarking, and monitoring post-sequencing QC metrics such as viral coverage, duplication rates, and base quality [95,96]. Well annotated bioinformatic pipelines with curated databases, efficient host-read filtration, and published algorithms are essential for accurate detection and cross-laboratory comparisons. Importantly, benchmarking studies highlight substantial variation in sensitivity, specificity, and error rates depending on the workflow choices and bioinformatic algorithms, reinforcing the need for external validation and standardised QC implementation throughout vmNGS-based surveillance [93,94,95,96,97].

## 3. Emerging Virus Discovery with vmNGS

Emerging zoonotic viruses have pandemic potential, as demonstrated by HIV, 2009/2010 IAV H1N1, and SARS-CoV-2, because they can spread through an immunologically naïve population who are often ill-prepared for countering the threat. Rapid identification of zoonotic viruses via passive surveillance of patients with suspected disease of unknown aetiology can accelerate the implementation of responses and tools to control spread. The lack of reference genomes and suitable targeted tests means vmNGS-based passive surveillance has been used to discover zoonotic viruses when they first spill into humans. Genomic surveillance is another critical aspect as monitoring mutations occurring in virus isolates facilitates epidemiological tracking of virus spread, identification of novel variants, and assessment of the continued effectiveness of currently used vaccines and antivirals based on mutations in immune epitopes and drug targets. An important consideration when interpreting virus discovery in clinical samples is that detection does not mean causation, and pathological or epidemiological investigation is often required to demonstrate pathogenicity [98,99].

### 3.1. The Central Role of vmNGS in COVID-19 Discovery and Response

SARS-CoV-2 was identified as the aetiological agent of the COVID-19 pandemic when it was first detected using vmNGS and PCR in the lower respiratory tract of 41 patients suffering from pneumonia in China in December 2019 [100]. Details of the vmNGS methodology used were not included in the initial publication from 24 January 2020 [100]. More details were included in a subsequent study on another of the first COVID-19 cases, a worker at the Wuhan Live Food Market with pneumonia, which was published on 3 February 2020 [101]. RNA extracted from the bronchiolar lavage fluid underwent rRNA depletion and untargeted vmNGS on the Illumina platform to generate over 50 million 150 bp reads [101]. Over 99% coverage and approximately 6X depth (number of reads calling base in each position) were obtained for the 29,903 bp genome, which was the first complete and annotated SARS-CoV-2 genome deposited on Genbank (a public database of genetic sequences) [101].

Phylogenetic and recombination breakpoint analyses of the obtained SARS-CoV-2 genome identified its closest known relatives and predicted it arose from recombination between bat sarbecoviruses, which, along with early COVID-19 cases coming from the Wuhan Live Food Market, suggested a zoonotic origin [101]. Inferences of the SARS-CoV-2 replication cycle were made possible by modelling its spike protein structure using the genetic sequence, which showed homology with the receptor-binding domain of SARS-CoV-1. This suggested SARS-CoV-2, like SARS-CoV-1, also used the ACE2 and TMPRSS2 as entry receptors, which was experimentally verified [101,102,103,104,105]. Having been used to identify early cases of SARS-CoV-2, complete genomes obtained by vmNGS were also used to infer phenotypic traits, which were often verified [101].

The value of publishing complete viral genomes is illustrated by the practical applications and research discoveries made possible by their availability through various sources including Genbank, Next Strain, and the Global Initiative on Sharing All Influenza Data, with the latter two used for phylogenetic analysis and genomic surveillance of SARS-CoV-2 and other viruses [106,107]. The first sequences were used to design a robust, specific, and sensitive PCR for the rapid diagnosis of SARS-CoV-2 infections, which was used to identify SARS-CoV-2-positive individuals who should quarantine as part of the lockdown protocols implemented by most countries [108,109]. Genetic sequences obtained early in the pandemic led to rapid vaccine development with 115 candidates already in the pipeline by 8 April 2020—just 3 months after the first SARS-CoV-2 sequences were made available [110,111]. Phase II/III clinical trials for vaccine candidates using more established technologies, like the CanSino adenovirus-vector vaccine, began in April 2020 [112,113], while trials for the more novel mRNA vaccines started in May 2020, with vaccine rollout occurring within a year of the first detection of SARS-CoV-2 [110,114,115,116]. Vaccines have been critical in preventing severe COVID-19 causing death or hospitalisation with active genomic surveillance used to monitor currently circulating variants and update vaccines to maintain their efficacy [117,118,119]. Most likely, the timeframe for control of COVID-19 was accelerated and the number of fatalities reduced due to discoveries made possible by the SARS-CoV-2 sequences obtained using vmNGS.

### 3.2. A Henipavirus Emerging in the Shadow of COVID-19

The power of vmNGS for unbiased virus discovery was clearly established in 2022, when the cause of an acute febrile illness in a patient in China remained undiagnosed using conventional methods. Using only vmNGS, a novel Henipavirus, subsequently named Langya virus, was identified as the aetiological agent [99]. Genetic information gained from vmNGS was used to design PCR tests for the screening of additional patients with similar unexplained acute febrile illness, resulting in the identification of 34 more cases [99,120]. Studies to identify potential animal reservoirs revealed high seropositivity in shrews (27%) and lower rates in dogs (5%) and goats (2%), suggesting these animals may have served as reservoir or intermediate hosts and highlighting potential sources of spillover and avenues for viral evolution [99,120]. While all identified cases were non-fatal and showed no evidence of human-to-human transmission [99,120], the precedent of highly pathogenic *Henipaviruses* such as Nipah and Hendra viruses underscores the need for close surveillance in line with the One Health approach [120,121].

## 4. Passive Surveillance of Reemerging Viruses Using vmNGS

Reemergence occurs when a pathogen previously under control undergoes a resurgence in case numbers, which normally involves one or a combination of the following: spread to previously unaffected locations; altered transmission dynamics; loss of population-level immunity; or cause of novel symptoms. Understanding and reacting to this phenomenon requires a One Health approach, because animal, environmental, human, and virological change can all give rise to reemergence. Targeted approaches are often unsuitable for reemerging virus detection, because there are often unexpected causes of disease and genetic changes driving virus reemergence, which could reduce the sensitivity of tests targeting the sites of mutation [122,123]. Untargeted vmNGS is, therefore, often used for initial identification of reemerging viruses at the start of outbreaks (Table 2). The genetic information obtained by vmNGS can identify mutations causing novel phenotypes resulting in virus reemergence, while NGS platforms are used for WGS during genomic surveillance of reemerging disease outbreaks.

### 4.1. vmNGS to Monitor Mpox Epidemics and Global Spread

A striking example is the recent reemergence and geographical spread of mpox clade IIb (formally monkeypox virus) in 2022/2023 and mpox clade Ib since late 2023 [129,130]. Factors contributing to mpox reemergence include waning cross-protective population-level immunity since the cessation of smallpox vaccination in 1980 and increased human–wildlife interactions driven by human behaviours [131].

Mpox clade IIb descended from mpox IIa, for which almost all human cases were from rural regions of West Africa and were contracted via zoonotic transmission [132,133]. This changed in 2017 when the ancestor of mpox IIb first emerged, causing an outbreak in urban areas of Nigeria, with HC-vmNGS used to obtain complete genome sequences for phylogenetic analysis, which demonstrated human–human transmission [134]. WGS of mpox isolated from travellers returning to the UK, Israel, and Singapore in 2018/2019 showed they had contracted the same virus causing the Nigerian outbreak [135,136,137,138].

The global spread of mpox IIb might have been kickstarted by travellers returning from Nigeria, which led to the 2022/2023 global health emergency as it spread to at least 110 countries [132,133]. vmNGS played an important role in monitoring its global spread, with the first complete mpox IIb genome obtained from a clinical sample by untargeted vm-NGS with the ONT platform in Portugal 2022 [139], and Illumina [140], ONT [141], and both [124] platforms were used to detect its introduction to previously mpox-free countries such as the Philippines and Brazil. Unlike the rare examples of suspected human–human mpox IIa transmission, which were mostly associated with direct or indirect contact with bodily fluids or skin pustules, a primary mode of mpox IIb transmission was sex between men, as shown by its high incidence in men who had sex with men, which had never previously been associated with an increased risk of mpox infections [132,133]. A vaccine rollout targeting this at-risk community was implemented to protect the gay and bisexual community and control the spread of mpox.

Mpox Ib descended from mpox clade Ia, which has historically caused more severe infections with a higher fatality rate than clade II mpox. Historically, clade I was a rare cause of disease in the Democratic Republic of Congo (DRC) and other countries into which the Congo Basin rainforest stretches [142]. The geographic confinement and rarity of clade I mpox changed in late 2023, when clade Ib emerged from the DRC and became endemic in the DRC, Burundi, and Uganda, along with outbreaks in previously unaffected African nations [142]. Vulnerable populations to mpox Ib infection include sex workers and children, indicating novel modes of transmission via heterosexual sex and close contact within households or schools, respectively [143]. The first complete mpox Ib genome was obtained using the Illumina platform for HC-vmNGS from a patient sample in the DRC in early 2024 [144], while vmNGS has continued to be used for WGS of DRC cases [145,146] and the first detected mpox Ib case in Europe [125].

The reemergence of mpox IIb and Ib was driven by the transition from primarily zoonotic to human–human transmission, with epidemiological evidence indicating that sexual transmission contributed to both [131]. This phenotypic change could be driven by the high number of GA-AA mutations observed in mpox IIb and Ib genomes compared to their ancestors [143,147]. GA-AA mutations are characteristic of APOBEC3 cytosine deaminase modifications introduced as part of the human antiviral immune response, suggesting undetected circulation in humans drove the evolution of mpox clades IIb and Ib [144,148]. WGS was key to uncovering potential genetic drivers of the mpox reemergence. These genomic sequences were also used to design specific PCRs for diagnosis, which also contributed to control efforts and identified the at-risk populations requiring vaccination [143]. Similar symptoms are caused by Herpesviruses and members of the *Orthopoxvirus* genus, which necessitates sensitive and highly specific molecular tests. These are designed by analysing complete viral genomes, while WGS of PCR positives is often used to verify results and facilitate epidemiological investigations [109,149,150,151]. vmNGS was important for mpox IIb detection in previously unaffected countries, while WGS contributed to technologies and studies that resulted in the containment of mpox IIb and the protection of gay and bisexual communities. Hopefully, pathogen identification and genomic surveillance using NGS will ensure mpox IIb remains under control and contribute to the response to contain mpox Ib outbreaks in Africa.

### 4.2. vmNGS in the Response and Understanding of Zika Virus Reemergence

Prior to 2013, Zika virus was just another mosquito-borne arbovirus (virus transmitted by arthropods), normally associated with asymptomatic, or rare cases of mildly symptomatic illness in Africa and Asia, but this changed when it caused Guillain–Barré syndrome during a little-publicised outbreak in French Polynesia [152]. The Brazilian epidemic marked a dramatic shift in public awareness of Zika virus, when its unexpected geographic expansion, association with microcephaly, and proof of vertical transmission was widely publicised one year after Brazil hosted the 2014 Football World Cup and one year before Rio de Janeiro hosted the 2016 Olympics [126].

Here too, vmNGS enabled direct identification of Zika virus from the amniotic fluids of infected pregnant mothers during the Brazilian outbreak [126]. To track the spread of Zika virus, the development of more rapid and sensitive diagnostics for screening less-severe cases of disease was a priority, so efforts were made to design PCR assays, with specificity being especially important due to the virus’ potential to cross-react with other Flavivirus species [153,154]. Zika virus RNA remained detectable by PCR significantly longer in urine (21 days post symptom onset) compared to serum (5 days) [155,156]. NGS also contributed to Zika virus genomic surveillance in the form of an amplicon-based WGS protocol, which was developed for Illumina and ONT platforms [157], while urine proved a suitable clinical sample for obtaining complete sequencing in some studies but not others [158,159]. As well as its role in surveillance, WGS also enabled comparison of the genetic sequences of Brazilian and pre-2015 isolates, which identified mutations linked to neurovirulence [160,161,162]. Increased retinoic acid response elements and phosphorylation sites in viral proteins were shown to reduce neurogenesis and increase apoptosis in neural progenitor cells [163,164].

Detection of Zika virus in mosquitos in 2014 and epidemiological investigations of viral genomes from the earliest known patients have shown it was probably introduced to Brazil in 2013, suggesting timely application of untargeted sequencing could have allowed for earlier intervention and potentially limited the scope of the epidemic [165,166]. Active surveillance of arthropods can be used to identify (re)emerging arboviruses, as outlined in Section 5.1.4, because infection of arthropods is a necessary stage of the infectious cycle that transmits arboviruses between different animal hosts.

### 4.3. vmNGS to Monitor the Spread and Reemergence of Arboviruses

Recently the threat posed by arboviruses has grown from a local to a global issue as climate change has expanded the geographic range of many arthropod vectors and the divide between the animals involved in sylvatic and urban cycles has crumbled as humans increasingly encroach on wildlife. With an estimated 400 million cases and 22,000 fatalities per annum, Dengue virus is the most prevalent arbovirus in humans, and its trajectory suggests cases numbers and geographic range will continue to grow [167]. Urbanisation and climate change have expanded the geographic range suitable for inhabitation by its main arthropod vectors, *Aedes aegypti* and *Aedes albopictus* [168], while globalisation has introduced these mosquitos to previously non-endemic countries [169,170,171]. The range of Dengue virus serotypes, which do not provide cross-reactive immunity, and the prevalence of asymptomatic infections hinders strategies to control its spread, while antibody-dependent enhancement makes vaccine design challenging [172,173,174]. The threat posed by Dengue virus, along with Oropouche virus (OROV), Mayaro virus, and Yellow Fever virus, has grown due to human activities in the Amazon—including deforestation, mining, and urban spread—which have intensified human interactions with the animals involved in sylvatic cycles, and resulted in subsequent increases in human cases [175].

Viral genetic factors have also contributed to arbovirus reemergence, such as the reassortment event that gave rise to the 2023/2024 OROV epidemic in Brazil and neighbouring countries [176,177,178]. The 2023/2024 virus had increased replication kinetics in mammalian cell lines compared to ancestral OROV, suggesting that the reassortment gave rise to a strain better-adapted to infect animal hosts [179]. Viral mutations can also give rise to changes in the virus–arthropod dynamics that increase arbovirus transmission, with the Chikungunya virus outbreak in 2005/2006 being an example of this [180]. Previously *Aedes aegypti* was its main vector, but amino acid changes in the glycoproteins of the Chikungunya virus envelope enhanced virus replication in *Aedes albopictus*, so it was more readily spread by this mosquito species [181,182,183]. The broader geographic distribution of *Aedes albopictus* compared to *Aedes aegypti* enabled global spread of Chikungunya, including autochthonous transmission in Europe [170,171,184,185].

Untargeted vmNGS of patient samples was used to detect genetic changes associated with the 2023/2024 OROV outbreak [178], while targeted, amplicon-based WGS has been used for genomic surveillance of arbovirus epidemics [186,187,188]. Identifying the serotype responsible for Dengue virus infections is vital due to its association with antibody-dependent enhancement, and thus a multiplexed, amplicon-based NGS method has been developed to reveal the serotype and obtain the complete genome sequence with a single test [189]. Point-of-care diagnostics are vital to strategies to control disease spread by rapidly identifying pathogens, and untargeted vmNGS was used for WGS of Dengue virus RNA recovered from positive antigen tests [190]. WGS is critical for phylogenetic characterisation of arboviruses, as is conducted during epidemiological studies, and can reveal genetic differences contributing to novel phenotypic traits contributing to their (re)emergence.

vmNGS has also been used in the diagnosis of arboviral disease. Arbovirus infections often cause acute febrile illness with non-specific symptoms, while multiple species typically circulate in the same location and are spread by the same vector host. This can hinder selection of suitable targeted diagnostic tests [123,191], while the reemergence of arboviruses in novel locations is another barrier to targeted-test selection [122,123,192]. When a syndromic approach to untargeted vmNGS surveillance of febrile patients in Uganda and Senegal was taken, Dengue, Rift Valley Fever, and Yellow Fever were all identified [193,194], and a novel *Orthobunyavirus* was discovered in a separate study from Uganda [195]. Co-infections by multiple arbovirus species can also occur, which can be detected using vmNGS [191,196]. There is evidence untargeted vmNGS has applications for the passive surveillance of potential arboviral diseases, which provides additional advantages of obtaining genetic information.

## 5. Enhancing (Re)Emerging Virus Surveillance with vmNGS

Surveillance using mNGS of animals and environments generates knowledge of the infectious threats that could be incorporated into active surveillance strategies (Table 3) [197]. The previous sections described the effective use of passive surveillance for (re)emerging virus detection post zoonosis, but there are fewer success stories for the proactive detection of high-risk viruses before they reemerge in human or other animal populations. Targeted detection methods remain the most cost-effective and sensitive approach to active pathogen surveillance, but the benefits of building awareness of the unknown and reducing costs of NGS make its use in active surveillance important and increasingly practical.

### 5.1. Active and Passive (Re)Emerging Virus Surveillance in Animals

Active surveillance of animals for known or high-risk (re)emerging viruses is a One Health-based strategy to prevent and prepare for potential zoonoses before they threaten human and animal health. Zoonotic viruses typically have complex host dynamics with a mixture of reservoir, intermediate and dead-end hosts. Reservoirs hosts are typically asymptomatic carriers of the virus, which amplify and transfer the virus to new locations. Animals found at the human–animal interface and in which viruses adapt to humans are intermediate hosts [198]. Dead-end hosts cannot transfer the virus to other individuals in their population or to other species but can suffer symptomatic disease. Genetic monitoring of known or high-risk zoonotic viruses in reservoir and intermediate hosts can identify mutations that pre-dispose viruses to zoonotic spillover. Ideally, surveillance should cover other non-human animals because spillover of a virus to a new species can generate selective pressures driving the evolution of new traits such as adaptation to humans. For this reason, animals are often used as sentinels for tracking and monitoring known or high-risk zoonotic viruses as part of active surveillance efforts. While passive surveillance in response to suspected infectious disease of unknown aetiology in non-human animals will protect animal health in the short-term and generate knowledge that could protect human health in the long-term.

#### 5.1.1. Farm Animals: The Need for vmNGS Surveillance

The frequency of interactions between livestock and humans makes them a potential source of zoonoses in workplaces or live food markets, where spillover between non-human animals increases selective pressures driving antigenic drift, which could adapt a virus to infect atypical hosts and gives it opportunities for recombination or reassortment. One example is common cold Human betacoronavirus OC43, which spilled over from cattle and might have caused a pandemic in the late 19th century (Table 4) [199,200].

In 2011, dairy farms in central and western Europe observed an unusual syndrome in adult cattle and sheep, characterised by reduced milk yield, diarrhoea, fever, and spontaneous abortions and severe birth defects in offspring. Conventional diagnostic tests failed to reveal the cause, prompting the use of vmNGS. This unbiased approach successfully identified a previously unrecognised *Orthobunyavirus*, now named Schmallenberg virus (SBV), as the aetiological agent [201]. Transmitted by the *Culicoides* midge, SBV rapidly spread west across Germany, the Netherlands, France, Belgium, the UK, and Ireland between 2012 and 2014 [202], which garnered wide concern among veterinarians and public health officials who were wary of potential adaptation to humans. However, subsequent surveillance has found no evidence of human SBV infections [203,204]. WGS and epidemiological studies suggested a viral reassortment event between Sathuperi and Shamonda viruses gave rise to SBV [205], though its exact origin remains unresolved [206,207]. More recently, further evidence of genetic reassortment between Sathuperi and Shamonda viruses was gained following WGS of an orthobunyavirus isolated from cattle in Japan 2022 with the same genetic makeup as SBV [208]. SBV emergence is a prime example of the role vmNGS can play in One Health surveillance and virus discovery at the human–animal interface.

**Table 4 ijms-26-09831-t004:** Known zoonotic viruses of livestock animals. ^1^ Shedding of Japanese encephalitis virus and swine–swine transmission has been demonstrated, with swine considered intermediate hosts that transmit the virus to mosquitos then onto humans.

Livestock	Zoonotic Virus	Location	Year	References
Cattle	Bovine coronavirus	Russia	Pre-1889	[199,200,209]
Swine	Influenza virus H1N1	Mexico	2009	[210]
	Hepatitis E Virus	Multiple (e.g., USA)	Multiple (e.g., 1998)	[211]
	Nipah virus	Malaysia and Singapore	1998	[212]
	Japanese encephalitis virus ^1^	Unknown	Unknown	[213]
Poultry	Avian influenza A virus	USA or France	Pre-1918	[210,214]
	Newcastle Disease virus	USA	1965	[215]
	West Nile virus	Israel	1998	[216]
Horses	Hendra virus	Australia	1994	[217]

Animals farmed for their fur or medicine, including mink, foxes, rabbits, and certain rodents, represent an important, often overlooked interface for zoonotic emergence. These species are not only susceptible to traditional animal viruses but have shown vulnerability to reverse zoonoses, acquiring infection from humans. Notably, mink have become infected with SARS-CoV-2 [109,218] and Hepatitis E virus [219], with outbreaks documented worldwide. In addition, HPAIV has been isolated from mink populations [220], raising concerns that they could serve as a reservoir, or even mixing vessels, where viral reassortment could occur. vmNGS has enabled comprehensive mapping of their virome, uncovering both known and unexpected viruses with potential zoonotic risks [48]. Mapping the virome of 461 small mammals farmed for fur or traditional medicines in China between 2021 and 2024, including foxes, mink, rabbits, and rodents, revealed the presence of multiple high-concern virus families (*Coronaviridae*, *Flaviviridae*, *Hepeviridae*, *Orthomyxoviridae,* and *Paramyxoviridae*), and detected known zoonotic viruses such as Japanese encephalitis virus and Hepatitis E virus [48]. The detection of such viral diversity in a relatively small sample set underscores the global risk posed by fur farming. Active surveillance using mNGS is a powerful approach to monitor and pre-empt zoonotic spillover from this unique animal–human interface.

#### 5.1.2. Wildlife Reservoirs and Metagenomic Surveillance: Preventing Zoonotic Spillover

Wild species, particular those in the orders Carnivora and Rodentia, are key reservoirs for zoonotic viruses and are often used as sentinels to assess spillover risk [49,221,222,223,224,225,226]. Although direct human–wildlife contact is generally lower than for farmed animals, episodic events, such as natural disasters, or human activities, such as deforestation, can disrupt ecological boundaries and raise the risk of zoonotic spillover.

Bats exemplify the critical role of wildlife in zoonosis, having been identified as original reservoirs for high-impact viruses such as Ebola virus in West Africa [225], Nipah virus in Southeast Asia [212], and sarbecoviruses in China [226]. Global vmNGS studies mapping bat viromes have uncovered a wide array of potential zoonotic viruses including members of *Bunyaviridae*, *Coronaviridae*, *Hantaviridae,* and *Picornaviridae* [227,228,229,230,231]. Data mining and modelling have been used to identify the geographic hot spots—predominantly in Africa, South America, Southeast Asia, Subcontinental Asia, and Eastern China—and animal taxa with the highest zoonotic risk [232,233,234], which can be used to guide resource allocation and sampling strategies for maximum impact.

#### 5.1.3. Companion Animals: A Critical Interface for One Health Surveillance

Companion animals, dogs and cats, predominantly, occupy a unique position at the human–animal interface due to their ubiquitous presence in households and the frequency of close physical contact with their owners. The high level of attentiveness to companion animals makes them valuable candidates for both passive and active surveillance of (re)emerging viruses. Notable examples of zoonotic pathogens include Rabies virus and Norovirus from dogs to humans with canine rabies consistently monitored via the integration of WGS and NGS into PCR-based surveillance programmes [235,236]. Zoonotic transmission from cats is comparatively rare, with occasional records of rabies infections following feline bites [237]. Cats are highly susceptible to often fatal IAV infections [238] and can be infected by SARS-CoV-2 via reverse-zoonosis [239].

Recognition of the role dogs and cats can play in zoonotic transmission is growing, with vmNGS having been used to map their viromes [240,241]. A study from 2022/2023 in China mapped the oralpharyngeal and rectal microbiomes and viromes of diseased and healthy cats and dogs, which identified animal pathogens not previously associated with disease and some zoonotic viruses and bacteria [240]. Another study mapping the gut virome of healthy dogs in China also showed they harbour known zoonotic viruses with alphacoronaviruses closely related to human common cold coronaviruses identified in both studies [240,241]. Awareness of the viruses harboured by companion animals is required to assess the risk they pose as sources of zoonotic pathogens.

#### 5.1.4. Expanding the Scope of Arbovirus Surveillance

Arthropod vectors transmit arboviruses, with mosquitoes being the most widely recognised group. However, a wide array of insects including midges, ticks, flies, fleas, and lice also contribute to the spread of vector-borne pathogens at the human–animal–environmental interface. Mapping the arthropod virome using vmNGS identifies arboviruses most likely to cause disease in humans and enable their genomic surveillance, which is critical, as many arboviral diseases are similar, and their geographic ranges overlap [242,243,244,245]. The importance of knowing which Dengue virus serovars are circulating was described in Section 4.3, with their early identification possible through genomic surveillance of their mosquito vectors [246,247,248,249]. Applications of vmNGS for screening arthropod vectors include gaining knowledge of arboviruses circulating in a specific location with potential to spill into humans [250,251]; identification of arthropod and viral species diversity [252,253,254]; and an understanding of the temporal and geographical dynamics influencing arbovirus circulation in insect populations [254]. The richness of arbovirus diversity, their spread to new locations, and their expanding seasonality hinder the selection of suitable targeted tests, so vmNGS is an attractive alternative for their active surveillance.

### 5.2. Influenza: A Recurring Issue

There have been four pandemics caused by influenza A viruses (IAV) since 1918, making it the only virus species to cause more than one pandemic in the 20th and 21st centuries [255]. They all emerged in human populations following zoonotic spillover from swine and avian species [210,256,257]. Considering this, public health officials remain vigilant to the risk of HPAIV—responsible for the ongoing panzootic of birds and marine mammals—adapting to efficiently infect humans. Passive surveillance strategies for HPAIV in at-risk animals such as birds of prey and cattle have been implemented in the USA with PCR screens for initial detection followed by WGS using amplicon-based or HC-vmNGS [258,259]. Active surveillance of HPAIV has also been conducted using environmental faecal samples and hunted ducks in Italy 2022–2024, which screened samples by PCR and carried out WGS to identify 5.26% positivity from 3497 samples [260]. Influenza virus has also been detected following environmental surveillance of wastewater, pointing to another avenue for active surveillance [261,262,263].

Rapidly evolving RNA viruses such as influenza require active genomic surveillance to monitor potential immune escape mutations and update vaccines to maintain their effectiveness. The Global Influenza Surveillance and Response System monitors circulating influenza to guide vaccine design [264]. Technological advances are facilitating the rapid generation of updated vaccines in response to the emergence of novel influenza strains [265].

Beyond surveillance, vmNGS has already been used in IAV detection with a retrospective detection of the 2009/2010 pandemic IAV H1N1 in 17/17 human infections following spillover from swine [14], while Illumina and ONT technologies have both been used to detect HPAIV in humans [266]. HPAIV WGS has been used to monitor mutations associated with human adaption with some arising in mammalian hosts [220,267]. Although sustained human–human HPAIV transmission has not yet been observed, surveillance of its multiple animal hosts is necessary to monitor zoonotic risk, with vmNGS being a useful tool should it spill into an unexpected host and for complete genome sequencing for epidemiological studies [262,268].

### 5.3. Rooting Through the Rubbish: Wastewater Surveillance

The principle behind environmental pathogen surveillance stems from infected hosts shedding virus particles and genetic material into their environments. As most environments are shared by multiple organisms, this approach enables the routine active surveillance of mixed populations with a single sample. Wastewater is the most studied environmental matrix for early detection and monitoring of reemerging viruses or ongoing outbreaks with SARS-CoV-2, IAV, and mpox detected using HC- and PEG-vmNGS [68,69,70,71,269,270]. Additional reemerging or high-risk zoonotic viruses detected by HC- or PEG-vmNGS of wastewater include Hepatitis E virus [71,271]; Chikungunya virus, Jingmen tick virus; Rabies virus [70]; and species of *Alphavirus*, *Flavivirus,* and *Betacoronavirus* [71]. From being on the cusp of eradication, polio virus is now reemerging, so its eradication programme is being stepped up. Wastewater surveillance is becoming an increasingly used component of the polio virus eradication programme with retrospective studies detecting polio virus in the wastewater of regions effected by polio, including in New York [72] and Israel in 2022/2023 [272]. Polio was also detected in London wastewater in 2022 before any cases were reported, which led to a vaccination drive among children in London with no cases subsequently reported, suggesting pathogen detection in wastewater can guide public health officials in real-world situations [273]. Opinions differ on whether environmental wastewater surveillance can proactively detect pathogens and if mNGS is sufficiently specific and sensitive, so it appears further testing of its role from the academic sector is necessary to fully ascertain its effectiveness.

## 6. Discussion

The literature on vmNGS reveals a diverse array of targeted and untargeted workflows, and with no single approach meeting the needs of all sample types, surveillance goals, or resource settings. Section 3 of this review outlined the tools available for vmNGS workflows, from sample preparation choices and host depletion or viral enrichment strategies to sequencing platforms and bioinformatic pipelines, empowering stakeholders to select workflows best suited to their clinical or environmental surveillance objectives. Given the rapid pace of innovation in NGS technologies, periodic reassessment of these options will remain essential.

Current applications demonstrate that the workflows described in Section 3 have already delivered substantial returns in both the discovery and surveillance of (re)emerging viruses. While protocols differ between studies, consistent trends have emerged: HC- or PEG-based enrichment is often favoured for wastewater-based environmental surveillance [68,69,70,71,269,270], while untargeted approaches dominate in clinical settings, especially for diagnostics of unusual or unexplained cases such as SARS-CoV-2 [100,101] and Schmallenberg virus [201]. Even untargeted approaches frequently incorporate some level of host depletion, most commonly rRNA removal to enhance viral signal.

Passive and syndromic surveillance has been the most impactful use of vmNGS to date, enabling rapid detection and characterisation of novel threats like SARS-CoV-2 [100,101] and reemerging viruses like mpox clades IIb/Ib [124,125] (Figure 5A). In such scenarios, vmNGS is deployed when there are strong imperatives, such as public health urgency, that justify its higher cost, and its turnaround is markedly faster than alternative untargeted tools such as electron microscopy. Critically, the genomic data generated by vmNGS has informed the design of diagnostics and vaccines [108,110,111], and strengthened outbreak response by revealing viral evolution, transmission patterns and population vulnerability [132,133,143]. Detection of potential immune escape mutants through active genomic surveillance enables vaccines to be updated periodically in line with currently circulating variants [117,118,119,264,265].

Incorporating vmNGS into ongoing genomic surveillance provide additional value during outbreaks. WGS data facilitates high-resolution genomic surveillance enabling public health agencies to track viral adaptation, origin and dissemination routes identifying mutations causing phenotypic changes, such as immune evasion, altered virulence profiles, or novel transmission routes, resulting in virus reemergence [143,144,147,148,160,161,162,178,181]. Advances in sequencing platforms coupled with improved decision-making on when and where to deploy vmNGS could further shorten detection times, increase coverage and accuracy, and reduce costs.

By contrast, adoption of vmNGS into active surveillance remains comparatively rare. The expense of regularly screening large numbers of samples, combined with the need for specialised analytical expertise, constrains its routine use. Where it has been employed, semi-targeted vmNGS approaches, with HC- or PEG-, have proven effective in wastewater monitoring [68,69,70,71,269,270], while active surveillance of sentinel animals, such as arthropods, bats and small mammals farmed for fur or medicine, has identified potential zoonotic threats before they were detected in humans [48,222,223,228]. These findings can inform the targeting of more cost effective and strategic active surveillance programmes (Figure 5B). However, for routine active surveillance to become feasible, investment in local sequencing capacity, cost reduction, automated workflows and clear decision-making frameworks will be essential.

The COVID-19 pandemic exposed stark global inequalities in NGS capacity, with high-income countries (HICs) producing 100-fold more viral genome sequences than low- and middle-income countries (LMIC) [274]. Although infectious disease burdens are often highest in LMICs, substantial barriers remain for the implementation and sustainability of vmNGS, [274,275,276] including major upfront cost for sequencing platforms, reagents computational infrastructure and workforce training [274,275,276,277]. While some LMICs acquired their sequencing infrastructure during the pandemic, ongoing obstacles such as high costs for consumables and access to computational resources hinders the sustainability of mNGS programmes [274,275,276,277]. Sporadic and reactive mNGS-based passive or active surveillance has been conducted in various LMIC [126,144,145,178,193,194,195], but challenges around the sustainability of this approach and an over reliance on foreign funding and expertise hinders its routine adoption in national public health frameworks.

Ethical considerations are closely intertwined with these capacity limitations, especially for LMIC. Equitable access to the benefits of vmNGS, data sovereignty, and fair data-sharing practices remain critical issues. The Nagoya protocol initiated in 2014, aims to address inequities stemming from the international use of genomic resources, but concerns persist regarding the privacy of clinical data and the adequacy of consent in both clinical and environmental surveillance [278]. Ensuring that surveillance activities respect local and global privacy standards as well as complying with directives such as the EU Habitat Directive, 92/43/EEC, for environmental sampling, remains essential to protect both affected communities and ecosystems [279,280].

Addressing these limitations and ethical challenges will require ongoing international collaboration, investment in sustainable local capacity, harmonised regulatory frameworks, and policies that promote equitable access and benefit sharing.

## 7. Conclusions

vmNGS offers a diverse tool kit that must be thoughtfully incorporated into workflows with careful attention to each step—sample selection and processing, host nucleic acid depletion and virus enrichment, sequencing and bioinformatic analysis—to ensure sensitive and specific viral detection. Workflow design should consider sample type (environmental or clinical), viral genome integrity, and contamination risks to maximise performance.

vmNGS has already proven valuable in passive surveillance, rapidly detecting (re)emerging viruses, especially for the discovery of unknown pathogens where targeted diagnostics are insufficient. Deployment of vmNGS for WGS has not only expedited vaccines and diagnostics development but also enabled real-time tracking of outbreak dynamics, to inform public health responses.

However, active surveillance applications remain relatively limited largely due to the costs and logistics of routine mass screening. The broader and more impactful use of vmNGS in both passive and active surveillance will depend on continued technological innovation, improved affordability, streamlining of workflows and more strategic decision-making around where and when to deploy the approach.

## 8. Future Directions

Progress in this field will depend on sustained collaboration between academia, industry and public health authorities to both drive the development of NGS and rigorously evaluate their impact within real-world surveillance frameworks. Surveillance is a key component of One Health approaches to pandemic preparedness with the rising threat of zoonotic and arboviral diseases demanding international and interdisciplinary cooperation and capacity building, particularly in resource-limited and rapidly changing environments. Community and stakeholder engagement is crucial to ensure surveillance strategies are contextually appropriate and equitable, while effective policy implementation by governments will be needed to mainstream and sustain these innovations.

By fully embracing a One Health perspective, future surveillance strategies powered by advanced sequencing technologies can facilitate early detection of (re)emerging and novel pathogens and enable more effective responses strengthening our collective ability to prevent or contain the next pandemic.

## Figures and Tables

**Figure 1 ijms-26-09831-f001:**
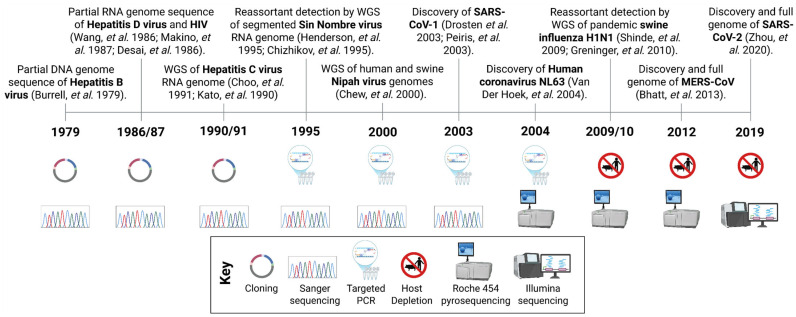
Milestones in virus genome sequencing technologies. The timeline illustrates the chronological progression of sequencing technologies and their impact on the discovery and characterisation of human viruses from 1979 to 2019. The evolution also emphasises how technological advances have transformed viral discovery and molecular epidemiology, enhancing surveillance capabilities to enable more rapid responses to (re)emerging threats [2,3,4,5,6,7,8,9,10,11,12,13,14,15,16,17]. Abbreviations: WGS, whole genome sequencing. Created in Biorender (biorender.com).

**Figure 2 ijms-26-09831-f002:**
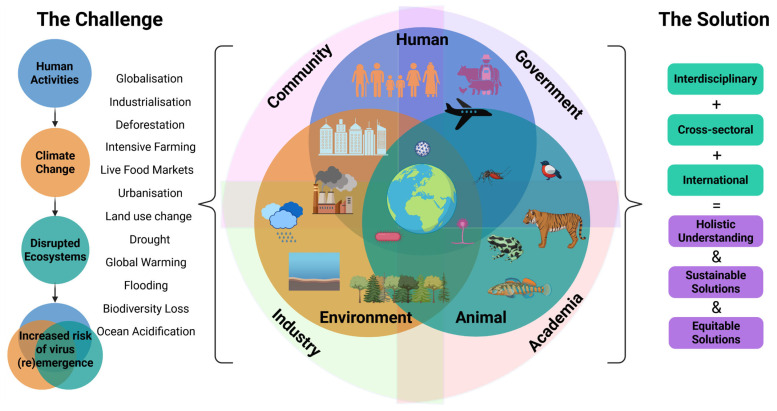
The One Health paradigm in (re)emerging virus investigation and control. The One Health paradigm recognises the interconnectedness of animal, environmental, and human health, promoting a holistic understanding of global health challenges and the pursuit of sustainable and equitable solutions with long-term benefits. This approach is particularly critical in the investigation and management of (re)emerging pathogens, as viral spillover events occur at the interface between these domains. Multiple ecological, biological, and socio-economic factors influence spillover risk, and the resulting impacts span the public health, veterinary, agricultural, and environmental sectors, making integration of a One Health perspective essential for effective research, prevention and control of (re)emerging viral threats. Created in Biorender (biorender.com).

**Figure 3 ijms-26-09831-f003:**
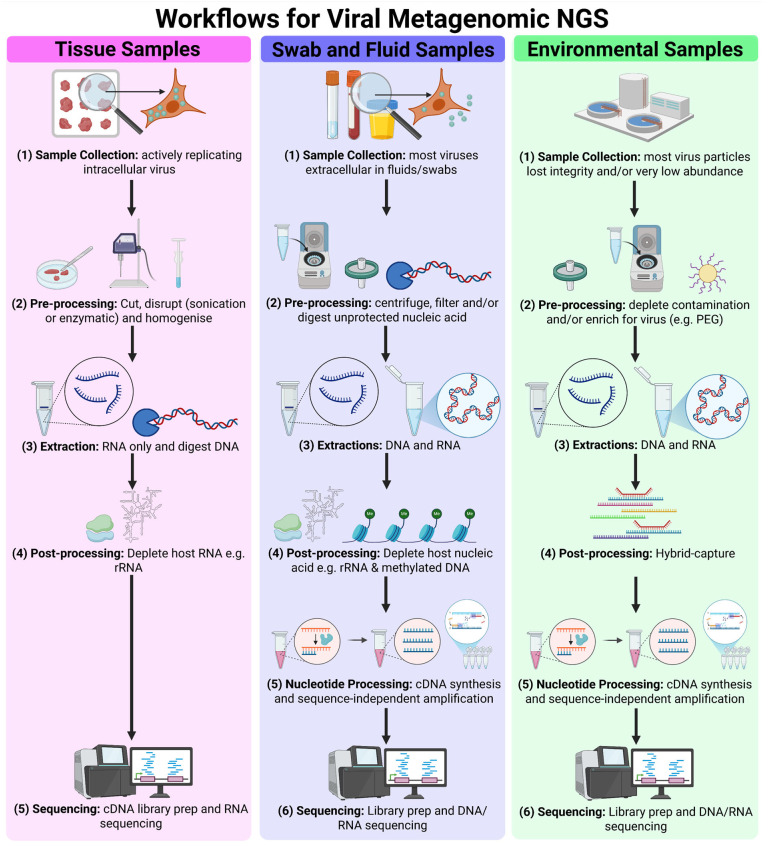
Workflow adaptations for different sample matrices in viral metagenomic NGS (vmNGS). Illustrated workflows show how vmNGS sample processing can be adapted based on the type of sample matrix, technological advancements, and available resources. For tissue samples, RNA extraction is typically prioritised, as actively replicating intracellular viruses are expected to produce viral transcripts. For swabs, biological fluids, and environmental samples, both DNA and RNA should be analysed, since viruses are often extracellular and may lack active transcription. Sequence-independent PCR amplification is often applied to extracts with low nucleic acid concentrations, particularly from swabs, fluids, and environmental matrices. For environmental samples, viral particles may have lost capsid integrity, leaving nucleic acids unprotected; in such cases, nuclease digestion to remove host nucleic acids is not recommended. Created in Biorender (biorender.com).

**Figure 4 ijms-26-09831-f004:**
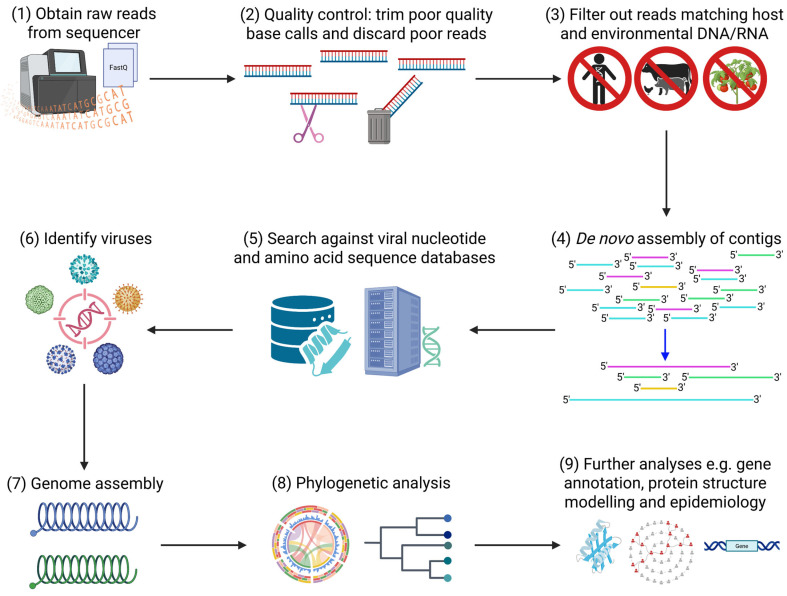
Bioinformatics pipelines for viral metagenomic NGS. vmNGS bioinformatics pipelines begin with raw sequence reads produced by NGS instruments, which require quality control steps to trim poor quality base calls, discard poor quality reads, and filter out host reads. Overlapping reads then undergo de novo assembly to generate contigs. Nucleotide sequences and, following in silico translation of the three open reading frames, amino acid sequences are searched against viral nucleotide and amino acid databases to identify detected viruses. Reference genomes can be used to assemble whole or more complete genome sequences, which can be used for phylogenetic analysis, epidemiology, and functional characterisation of viruses. Created in Biorender (biorender.com).

**Figure 5 ijms-26-09831-f005:**
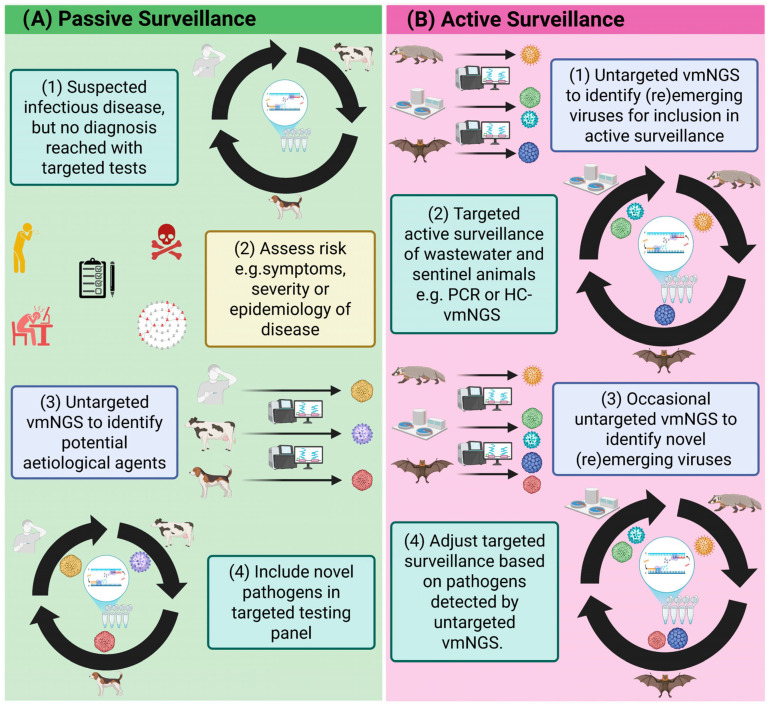
Integration of vmNGS into passive (**A**) and active (**B**) surveillance. (**A**) Passive surveillance involves reacting to a disease outbreak. vmNGS should be deployed when infectious disease is suspected, aetiological agents are unknown, and risk of a larger or more severe outbreak is high. Targeted methods can then be designed to more cost-effectively and rapidly detect novel viruses. (**B**) Active surveillance proactively screens high-risk animals and environments for potential (re)emerging pathogens before they cause an outbreak. The high number of samples required makes vmNGS too inefficient and expensive to be used routinely. It can be deployed when scoping out the viruses that should be included in active surveillance of a new region or to periodically update the viruses requiring inclusion. Created in Biorender (biorender.com).

**Table 1 ijms-26-09831-t001:** Practical and technical overview of sequencing technologies [19,20,21]. ^1^ Cost per base does not include initial costs for instrument purchase. ^2^ Adaptive sequencing enables the real-time selection of target sequences or exclusion of off-target sequences. Abbreviations: ONT, Oxford Nanopore Technologies; ds-cDNA, double stranded complementary DNA; SNP, single nucleotide polymorphism; WGS, Whole genome sequencing.

Generation	First (Sanger)	Second (Illumina)	Third (ONT)
Other Platforms	Maxam-Gilbert	Roche 454, Ion Torrent, SOLiD	PacBio
Cost per Kb ($) ^1^	500–1000	0.01–0.10	0.10–10.00
Error Rate (%)	0.001	0.1–1.0	1–15
Output (bases per run)	1000 bp	60 Gb–6 Tb	10 Gb–10 Tb
Read Length	1000 bp	50–300 bp	Up to 1+ Mb
Requirement for PCR	Yes (sequence-dependent to enrich target)	Yes (sequence-independent for cluster generation on flow cell, can also be used in library preparation and to enrich targets)	No (can be used to enrich targets)
Preparation Methods	PCR or clonal amplification of target	Generate library: fragmentation, ds-cDNA synthesis, adapter/barcode ligation and purification	Generate library: adapter/barcode ligation and purification
Data Storage Requirements	Low	High	High
Portable	No	No	Yes
Suitable for Metagenomics	No	Yes (including degraded samples)	Yes (unsuitable for degraded samples)
Other Applications	Genotyping and Targeted Sequencing	WGS, SNP Variant Calling and Transcriptomics	WGS, Splice Variant Detection, Real-time Sequencing, Direct RNA Sequencing, Epigenetic Modification Detection and Adaptive Sequencing ^2^
Main Strengths	High sensitivity (for target only), high specificity and high accuracy	High accuracy, high sensitivity, high depth, high throughput and suitability for degraded samples	Long read length, high throughput, high sensitivity and high coverage
Main Weakness	Low throughput	Short read-length	Poor accuracy

**Table 2 ijms-26-09831-t002:** Overview of reemerging viruses where vmNGS contributed to their identification and characterisation. Abbreviations: DRC, Democratic Republic of Congo.

Virus	Original Location	New Location	Year	Cause	Refs.
Mpox IIb	Nigeria	Worldwide	2022	See main text	[124]
Mpox Ib	DRC	Worldwide	2023	See main text	[125]
Zika virus	Africa, Asia and French Polynesia	Brazil	2015	See main text	[126]
Oropouche virus	Brazil, Caribbean, Peru and Panama	Ecuador	2016	Unknown, PCR recognised mutated region	[122,123]
St Louis Encephalitis virus	Argentina	USA	2015	Migratory birds	[127]
Ebola virus	DRC	Guinea, Liberia, Sierra Leone	2014	Zoonotic spillover, probably bats	[128]

**Table 3 ijms-26-09831-t003:** Recommended vmNGS approaches by surveillance objective and sample Type. For all surveillance objectives, method selection should balance sample integrity, cost, and laboratory capacity. ^1^ Illumina shotgun sequencing is preferred for degraded or complex samples due to higher sequence accuracy. ONT enables rapid sequencing for high-integrity samples or when longer reads are beneficial.

Surveillance Objective	Sample Type	NGS Methods
Active or passive surveillance for (re)emerging respiratory disease agents	Nasopharyngeal swabs, lung biopsy (postmortem only)	Untargeted ^1^ or semi-targeted vmNGS
Active or passive surveillance for (re)emerging gastrointestinal disease agents.	Cloacal swabs, faecal swabs/samples, gut biopsy	Untargeted ^1^ or semi-targeted vmNGS
Active or passive surveillance for neurological disease agents	Cerebral spinal fluid (CSF) or brain biopsy (postmortem)	Untargeted ^1^ or semi-targeted vmNGS
Surveillance of undifferentiated illnesses or sentinel animals	Spleen biopsy or blood	Untargeted ^1^ or semi-targeted vmNGS
Environmental surveillance, early outbreak detection	Wastewater, farm effluents, soils	Semi-targeted vmNGS

## Data Availability

No new data were created or analyzed in this study. Data sharing is not applicable to this article.

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
