# Peer review of "Viral Metagenomic Next-Generation Sequencing for One Health Discovery and Surveillance of (Re)Emerging Viruses: A Deep Review"

_ijms, 2025, doi:10.3390/ijms26199831_

Round 1

Reviewer 1 Report

Comments and Suggestions for Authors

Thanks for the opportunity to review the manuscript by Russell et al. In this review, the authors provide a closer examination of the strengths and weaknesses of vmNGS in both passive and active surveillance. It addresses issues such as cost, infrastructure needs, and the importance of teamwork across various disciplines. By combining molecular, ecological, and public health views, vmNGS becomes a key tool for early detection, thorough monitoring, and informed action against new and resurging viral threats. This highlights its crucial role in preparing for global pandemics and controlling diseases that spread from animals to humans.

The manuscript is quite extensive and addresses a variety of essential themes in the field in depth.

These figures are beautifully designed, and the graphics used are top-notch. The tables and boxes also do a great job of summarizing the information in a clear and practical way.

I really do not have any critical comments for the authors. My only suggestion might be to add ''a deep review'' at the end of the title.

Author Response

We appreciate your recommendation to update the manuscript title. The change has been made and is highlighted in the revised version.

Reviewer 2 Report

Comments and Suggestions for Authors

The manuscript "Viral Metagenomic Next-Generation Sequencing for One Health Discovery and Surveillance of (Re)Emerging Viruses", is an extensive and well-written review that covers several important aspects of the use of vmNGS for surveillance, mainly of zoonotic viruses.

Although at first I thought it was a lengthy review, I must say that it thoroughly covers the topic and effectively highlights both the importance of NGS technologies in virus surveillance and the success of prevention and control efforts using these approaches.

It is a very nice review

Author Response

Thank you for raising concerns regarding the length of the review. We share your view and have endeavoured to limit further extension of the manuscript while ensuring all feedback is thoroughly addressed.

Reviewer 3 Report

Comments and Suggestions for Authors

In this review article, Russell et al., discuss the importance of viral metagenomic next generation sequencing for untargeted detection, as well as characterization of the emerging zoonotic viruses by utilizing active as well as passive surveillance methods

Overall, a well written and researched review article . I did however find a few sections with either typos, formatting issues, or where the text could be improved:

  1. Please format table 1 such that there is some space between it and the text that follows
  2. Section 2 can be reorganized such that the text about environmental section in 2.1.1, and the text about clinical samples, including tissue, swabs and fluid samples can stay in section 2.1.2, but the quality control section which originally was in section 2.1.3 should perhaps be 2.2. I believe sample QC and sample origin are and should be presented as distinct things
  3. There are a lot of acronyms used throughout the review. Please make sure to include a list of abbreviations at the end of the manuscript.
  4. Please ensure that text in section 2.3, wherever applicable, refers back to points from table 1
  5. On page number 20, I believe table 3 has been mislabeled as table 1
  6. I also did not find references in the text citing back to table 3. Please include that wherever applicable.
  7. Seems like active surveillance can also be useful for strain identification with respect to annual flu as well as Covid booster shot production, for diseases which have caused multiple pandemics. It might be useful to mention this information in reference to vaccine development.

Author Response

We are grateful for your detailed comments.  All suggestions have been incorporated into the revised manuscript, and point-by point response are provided below for your reference while corresponding changes are marked in the document.

Comments 1: Please format table 1 such that there is some space between it and the text that follows

Response 1: A new line space (Line 79) has been added for Table 1 and Table 4 (formerly Table 3, Line 712).

Comments 2: Section 2 can be reorganized such that the text about environmental section in 2.1.1, and the text about clinical samples, including tissue, swabs and fluid samples can stay in section 2.1.2, but the quality control section which originally was in section 2.1.3 should perhaps be 2.2. I believe sample QC and sample origin are and should be presented as distinct things

Response 2: Section 2 has been re-numbered so quality control (Sample Integrity and Contamination) is its own section (2.5). This dedicated and expanded section named “Maintaining Quality Control and Standardisation of vmNGS” reflects the importance of QC, standardisation and performance metrics. Lines 381-418.

Comments 3: There are a lot of acronyms used throughout the review. Please make sure to include a list of abbreviations at the end of the manuscript.

Response 3: Abbreviations section 9 has been added. Lines: 963-990.

Comments 4: Please ensure that text in section 2.3, wherever applicable, refers back to points from table 1

Response 4: Table 1 has been referenced where appropriate in section 2.3. Lines: 308, 312, 320, 328 and 330.

Comments 5: On page number 20, I believe table 3 has been mislabeled as table 1

Response 5: This has been corrected to “Table 4”. Line 709.  

Comments 6: I also did not find references in the text citing back to table 3. Please include that wherever applicable

Response 6: Table 4 (formerly Table 3) has now been referenced in Line 688.

Comments 7: Seems like active surveillance can also be useful for strain identification with respect to annual flu as well as Covid booster shot production, for diseases which have caused multiple pandemics. It might be useful to mention this information in reference to vaccine development.

Response 7: The role of active genomic surveillance in vaccine design and effectiveness has been added for COVID-19 (Lines 471-473) and Influenza (Lines 792-797) and acknowledged in the discussion (Lines 853-855).

Vaccines have been critical in preventing severe COVID-19 causing death or hospitalization with active genomic surveillance used to monitor currently circulating variants and update vaccines to maintain their efficacy [1–3].

Rapidly evolving RNA viruses such as influenza require active genomic surveillance to monitor potential immune escape mutations and update vaccines to maintain their effectiveness. The Global Influenza Surveillance and Response System monitors circulating influenza to guide vaccine design [4]. Technological advances are facilitating the rapid generation of updated vaccines in response to the emergence of novel influenza strains [5].

Detection of potential immune escape mutants through active genomic surveillance enables vaccines to be updated periodically in line with currently circulating variants [1–5].

Reviewer 4 Report

Comments and Suggestions for Authors

1.The literature review lacks critical discussion on the limitations of vmNGS, particularly its feasibility in resource-limited settings.

2.Technical descriptions of vmNGS workflows need more precise details, such as sequencing depth and analytical parameters, to enhance reproducibility.

3.Claims regarding detection sensitivity, especially for low viral loads, are not consistently supported by robust empirical evidence across all sample types.

4.The review would benefit from including instances where vmNGS yielded ambiguous or unsuccessful results for a more balanced perspective.

5.Ethical considerations, including data sharing and privacy issues in environmental surveillance, are underexplored.

6.Computational requirements and bioinformatic challenges associated with vmNGS data analysis are inadequately addressed.

7.Statistical support for certain cited performance metrics, such as detection rates, is lacking, including confidence intervals or sample sizes.

8.A comparative table or flowchart recommending vmNGS methods based on sample type and surveillance objectives would enhance practical utility.

9.The conclusion should provide clearer guidelines for validating vmNGS findings using orthogonal methods.

10.The role of artificial intelligence in enhancing vmNGS bioinformatics warrants further discussion.

11.The manuscript should engage more deeply with standardization efforts and quality control frameworks across laboratory settings.

Author Response

Thank you for your insightful suggestions to improve the review. All comments have been carefully considered and addressed both in the main text and in the responses provided below. Where appropriate, related issues have been grouped together and responded collectively for clarity and completeness.

Comments 1: The literature review lacks critical discussion on the limitations of vmNGS, particularly its feasibility in resource-limited settings.

Comments 5: Ethical considerations, including data sharing and privacy issues in environmental surveillance, are underexplored.

Response to Comments 1 & 5: Comments 1 and 5 were addressed together, as both focus on substantial limitations and ethical concerns inherent to vmNGS. with the revised manuscript includes a dedicated section “Limitations and Concerns” in the Discussion. It discusses persistent challenges facing implementation and sustainability of mNGS  in resource-limited setting, such as inequities in access, infrastructure and capacity between low- and high-income regions. Moreover, ethical considerations pertaining to clinical and environmental surveillance applications of vmNGS are also covered. Reference to recent analyses on ethical considerations of mNGS in resource-limited settings has been incorporated. Lines 887-918. (References added to the article for the new Section 6.1. are in a bibliography at the end of this letter).

6.1. Limitations and Concerns

The COVID-19 pandemic exposed stark global inequalities in NGS capacity, with high-income countries (HICs) producing 100-fold more viral genome sequences than low- and middle-income countries (LMIC) [274]. Although infectious disease burdens are often highest in LMICs, substantial barriers remain for the implementation and sustainability of vmNGS, [274–276] including major upfront cost for sequencing platforms, reagents computational infrastructure and workforce training [274–277]. While some LMICs acquired their sequencing infrastructure during the pandemic, ongoing obstacles such as high costs for consumables and access to computational resources hinders the sustainability of mNGS programmes [274–277]. Sporadic and reactive mNGS-based passive or active surveillance has been conducted in various LMIC [118, 136, 137, 172, 187–189], but challenges around the sustainability of this approach and an over reliance on foreign funding and expertise hinders its routine adoption in national public health frameworks.

Ethical considerations are closely intertwined with these capacity limitations, especially for LMIC. Equitable access to the benefits of vmNGS, data sovereignty, and fair data-sharing practices remain critical issues. The Nagoya protocol initiated in 2014, aims to address inequities stemming from the international use of genomic resources, but concerns persist regarding the privacy of clinical data and the adequacy of consent in both clinical and environmental surveillance [278]. Ensuring that surveillance activities respect local and global privacy standards as well as complying with directive such as the EU Habitat Directive, 92/43/EEC for environmental sampling, remains essential to protect both affected communities and ecosystems [279, 280].

Addressing these limitations and ethical challenges will require ongoing international collaboration, investment in sustainable local capacity, harmonised regulatory frameworks, and policies that promote equitable access and benefit sharing.

Comments 2: Technical descriptions of vmNGS workflows need more precise details, such as sequencing depth and analytical parameters, to enhance reproducibility.

Comments 3: Claims regarding detection sensitivity, especially for low viral loads, are not consistently supported by robust empirical evidence across all sample types.

Comments 4: The review would benefit from including instances where vmNGS yielded ambiguous or unsuccessful results for a more balanced perspective.

Comments 6: Computational requirements and bioinformatic challenges associated with vmNGS data analysis are inadequately addressed.

Comments 7: Statistical support for certain cited performance metrics, such as detection rates, is lacking, including confidence intervals or sample sizes.

Comments 9: The conclusion should provide clearer guidelines for validating vmNGS findings using orthogonal methods.

Comments 11: The manuscript should engage more deeply with standardization efforts and quality control frameworks across laboratory settings.

Response to Comments 2, 3, 4, 6, 9 and 11: These comments have been addressed collectively via the addition of Section 2.5, which focuses on aspects of quality control and workflow standardisation. Grouping of the comments was based on the shared theme that for vmNGS to be incorporated reliably it requires improved analytical parameters/metrics, quality assurance and standardisation In the revised manuscript, section 2.5 details approaches to  standardisation including analytical parameters to assess workflow progress and sequence quality enabling inter-study comparisons, and orthogonal methods that can be used to validate and verify results as described below:

2.5. Maintaining Quality Control and Standardisation of vmNGS

QC is vital throughout vmNGS with standardisation of protocols and quality assurance metrics used for assessing workflow performance. Sample and nucleic acid integrity play critical roles in workflow selection and success. Highly degraded samples may limit the applicability of certain enrichment or sequencing approaches, such as host depletion methods (see section 2.2) that rely on intact viral particles and some long-read NGS technologies (see section 2.3). There are greater flexibility and options available when sample and nucleic acid integrity is maintained, which begins with strict adherence to standard operating procedures for collection, transport and storage (see Box 2).

Sequencing the metagenome can detect all nucleic acids, so contamination poses a significant risk to specificity. Contamination can arise externally from the laboratory environment, such as from personnel or sampling equipment; and internally, such as from cross-contamination between samples. While, complete avoidance of contamination is not possible, it can be minimised by implementing standard operating procedures (see Box 2) and tracked by including negative controls [6].

Consistent technical validity and comparability of vmNGS workflows rely on harmonised quality control (QC) metrics, robust proficiency testing, and precise documentation of technical parameters from sample acquisition through bioinformatic analysis. Unlike PCR, vmNGS lacks a targeted amplification step, making its LOD highly dependent on sequencing depth and sample integrity as only a small fraction of total reads may identify low-abundance viral genomes in complex matrices or degraded samples, thus requiring high coverage and increased costs [7–9]. Key determinants of sensitivity and reliability include flow cell capacity, careful selection and validation of sample and library quality, bioinformatic strategies, and the use of pooling strategies with unique molecular identifiers, which can improve efficiency but must be balanced against the risks of insufficient per-sample depth when batching multiple samples. A rigorous approach entails matching sample complexity and expected pathogen abundance to required depth, and validating vmNGS workflows with reference samples with known pathogen status or spiked in standards for benchmarking, and monitoring post-sequencing QC metrics such as viral coverage, duplication rates, and base quality [9, 10]. Well annotated bioinformatic pipelines with curated databases, efficient host-read filtration, and published algorithms are essential for accurate detection and cross-laboratory comparisons. Importantly, benchmarking studies highlight substantial variation in sensitivity, specificity and error rates depending on the workflow choices and bioinformatic algorithms, reinforcing the need for external validation and standardised QC implementation throughout vmNGS-based surveillance [7–11].

Comments 8: A comparative table or flowchart recommending vmNGS methods based on sample type and surveillance objectives would enhance practical utility.

Response 8: The authors agree this would enahnce the practical utility of the review.  Accordingly, we have incorporated a table presenting recommended combination of surveillance objectives, sample types and most appropriate NGS methods for each context. This table is referenced at the start of the section on surveillance of animals, where theuse of vmNGS in active surveillance is discussed, in Line 649, and the table legend is at Lines 658-662.  

Table 3. Recommended vmNGS Approches by Surveillance Objective and Sample Type. For all surveillance objectives, method selection should balance sample integrity, cost and laboratory capacity. 1) Illumina shot gun sequencing is preferred for degraded or complex samples due to higher sequence accuracy. ONT enables rapid sequencing for high integrity samples or when longer reads are beneficial.

Surveillance Objective

Sample Type

NGS Methods

Active or passive surveillance for (re)emerging respiratory disease agents

Nasopharyngeal swabs, lung biopsy (postmortem only)

Untargeted1 or

semi-targeted vmNGS

Active or passive surveillance for (re)emerging gastrointestinal disease agents.

Cloacal swabs, faecal swabs/samples, gut biopsy

Untargeted1 or

semi-targeted vmNGS

Active or passive surveillance for neurological disease agents

Cerebral spinal fluid (CSF) or brain biopsy (postmortem)

Untargeted1 or

semi-targeted vmNGS

Surveillance of undifferentiated illnesses or sentinel animals

Spleen biopsy or blood

Untargeted1 or

semi-targeted vmNGS

Environmental surveillance, early outbreak detection

Wastewater, farm effluents, soils

Semi-targeted vmNGS

Comments 10: The role of artificial intelligence in enhancing vmNGS bioinformatics warrants further discussion.

Response 10: The authors agree AI is a valuable tool for mNGS, which must be harnessed and the original draft neglected to discuss this in sufficient detail. So as not to extend the article too much the authors have given a very brief overview of the role of AI in mNGS with reference to a recent review we believe gives a very thorough and practical review of AI tools in Lines: 347-354.

A lack of suitable reference genomes for many newly discovered viruses and the sheer quantity of data generated by modern NGS instruments has transformed the bioinformatic landscape. As such, artificial intelligence (AI) tools are increasingly being developed and utilised for computationally intensive processes such as taxonomic identification, genome resolution and genome annotation [12]. Modern AI tools have already proven effective in mining existing mNGS datasets to expand the virosphere [13]. Limitations around sensitivity exist compared to more-established tools, with a thorough review of the advantages and disadvantages of AI-based tools published recently [12].

Round 2

Reviewer 4 Report

Comments and Suggestions for Authors

After my careful review, I found that the author has positively responded to all the points I am concerned about. The current version is acceptable.